# Selective Hydrogenation of Benzene to Cyclohexene over Ru-Zn Catalysts: Investigations on the Effect of Zn Content and ZrO$_2$ as the Support and Dispersant

**Haijie Sun** [1] , **Zhihao Chen** [2,*] , **Lingxia Chen** [1], **Huiji Li** [1], **Zhikun Peng** [3,*], **Zhongyi Liu** [3] **and Shouchang Liu** [3]

[1]  Institute of Environmental and Catalytic Engineering, College of Chemistry and Chemical Engineering, Zhengzhou Normal University, Zhengzhou 450044, Henan, China; sunhaijie406@163.com (H.S.); clingxia@vip.163.com (L.C.); huijili@zznu.edu.cn (H.L.)
[2]  Zhengzhou Tobacco Research Institute of CNTC, Zhengzhou 450001, Henan, China
[3]  College of Chemistry and Molecular Engineering, Zhengzhou University, Zhengzhou 450001, Henan, China; liuzhongyi406@163.com (Z.L.); liushouchang406@163.com (S.L.)
*  Correspondence: chenzh@ztri.com.cn (Z.C.); Zhikunpeng@163.com (Z.P.); Tel.: +86-371-6767-2762 (Z.C.); +86-158-3810-9080 (Z.P.)

**Abstract:** m-ZrO$_2$ (monoclinic phase) supported Ru-Zn catalysts and unsupported Ru-Zn catalysts were synthesized via the impregnation method and co-precipitation method, respectively. The catalytic activity and selectivity were evaluated for selective hydrogenation of benzene towards cyclohexene formation. Catalyst samples before and after catalytic experiments were thoroughly characterized via X-ray diffraction (XRD), X-ray Fluorescence (XRF), transmission electron microscopy (TEM), N$_2$-sorption, X-ray photoelectron spectroscopy (XPS), H$_2$-temperature programmed reduction (H$_2$-TPR), and a contact angle meter. It was found that Zn mainly existed as ZnO, and its content was increased in Ru-Zn/m-ZrO$_2$ by enhancing the Zn content during the preparation procedure. This results in the amount of formed (Zn(OH)$_2$)$_3$(ZnSO$_4$)(H$_2$O)$_3$ increasing and the catalyst becoming more hydrophilic. Therefore, Ru-Zn/m-ZrO$_2$ with adsorbed benzene would easily move from the oil phase into the aqueous phase, in which the synthesis of cyclohexene took place. The generated cyclohexene then went back into the oil phase, and the further hydrogenation of cyclohexene would be retarded because of the high hydrophilicity of Ru-Zn/m-ZrO$_2$. Hence, the selectivity towards cyclohexene formation over Ru-Zn/m-ZrO$_2$ improved by increasing the Zn content. When the theoretical molar ratio of Zn to Ru was 0.60, the highest cyclohexene yield of 60.9% was obtained over Ru-Zn (0.60)/m-ZrO$_2$. On the other hand, when m-ZrO$_2$ was utilized as the dispersant (i.e., employed as an additive during the reaction), the catalytic activity and selectivity towards cyclohexene synthesis over the unsupported Ru-Zn catalyst was lower than that achieved over the Ru-Zn catalyst with m-ZrO$_2$ as the support. This is mainly because the supported catalyst sample demonstrated superior dispersion of Ru, higher content of (Zn(OH)$_2$)$_3$(ZnSO$_4$)(H$_2$O)$_3$, and a stronger electronic effect between Ru and ZrO$_2$. The Ru-Zn(0.60)/m-ZrO$_2$ was reused 17 times without any regeneration, and no loss of catalytic activity and selectivity towards cyclohexene formation was observed.

**Keywords:** selective hydrogenation; benzene; cyclohexene; Ru; Zn; ZrO$_2$

## 1. Introduction

Selective hydrogenation of benzene towards cyclohexene synthesis has been a significant reaction in the field of catalysis research [1–5]. This is mainly attributed to the fact that the production of

caprolactam and adipic acid via cyclohexene is environmentally friendlier, more energy preservation, and might result in higher carbon atom economy in comparison to that via cyclohexane [6,7].

The first industrial plant for production of cyclohexene from selective hydrogenation of benzene over unsupported Ru-Zn catalyst was manufactured by Asahi in 1989 [8]. However, some drawbacks for the catalyst, such as high Ru content and the ease of being poisoned, are of great difficulty to overcome. Therefore, the development of supported Ru catalysts with relatively low Ru loading and high Ru dispersion have drawn great interest. Commonly, $ZrO_2$ is selected as a proper support. For instance, Zhou et al. [9] prepared Ru-Zn/$ZrO_2$ catalyst via the deposition-precipitation method, from which a 54% cyclohexene yield was obtained. Furthermore, Ru-Zn/$ZrO_2$ catalyst was prepared via a two-step impregnation method by Yan et al. [1], which gave a 48.5% cyclohexene yield. Moreover, Peng et al. [10] applied a chemical reduction method to synthesize a Ru/$m$-$ZrO_2$/$t$-$ZrO_2$ catalyst, from which a 55.3% cyclohexene yield was achieved. Liu et al. [11] also used the chemical reduction method to prepare a Ru-La-B/$ZrO_2$ catalyst, and a 53.2% cyclohexene yield was shown. Other than $ZrO_2$, zeolite (i.e., SBA-15) [12] and $\gamma$-$Al_2O_3$ [13] were also reported as the catalyst support for selective hydrogenation of benzene over Ru-based catalysts. However, how supports affect the catalytic activity and selectivity has been rarely addressed.

A promoter is one of the most effective ways to improve the cyclohexene yield over the Ru-based catalytic system. Zn [14,15], Fe [16,17], Co [18], Mn [19], La [20,21], and Ce [22,23] have been investigated as promoters for Ru catalysts on selective hydrogenation of benzene. Zn has been widely studied and reported among all promoters due to its superior promotion performance [15]. However, the status of Zn (e.g., valence of Zn) in supported Ru-based catalysts has been controversial. Zhou et al. [24] and Wang et al. [25] deemed that in Ru-Zn/$ZrO_2$, Zn existed as metallic Zn covering some of the Ru active sites, with lower selectivity towards cyclohexene formation. They also suggested that the Zn@Ru system could be achieved by doping metallic Zn into Ru, which spontaneously modified the geometric and electronic structure of Ru, and thus improved the catalytic activity and selectivity towards cyclohexene formation. On the other hand, Zhou et al. [9] and Yan et al. [1] demonstrated that Zn existed as ZnO in Ru-Zn/$ZrO_2$. However, the mechanism of ZnO affecting the formation of cyclohexene was not clearly explained. Thus, it is of great significance to investigate the status of Zn in Ru-Zn/$ZrO_2$ and how it affects the catalytic system, which could provide essential guidance for the development of the supported Ru-based catalysts.

Based on the previous work, we prepared Ru-Zn/$m$-$ZrO_2$ catalysts with different Zn content via the impregnation method. In order to reveal the status of Zn in Ru-Zn/$ZrO_2$ and how it affects the catalytic system, all samples were evaluated for selective hydrogenation of benzene towards cyclohexene formation. In addition, unsupported Ru-Zn catalyst with the same content of Ru and Zn was synthesized via the co-precipitation method, which was tested under the same experimental conditions by adding $ZrO_2$ as a dispersant. In comparison to that observed over Ru-Zn/$ZrO_2$, the support effect of $ZrO_2$ was proposed for the catalytic performance of Ru-Zn on the selective hydrogenation of benzene towards cyclohexene generation.

## 2. Results

### 2.1. Effect of Zn Content

XRD patterns of Ru-Zn(x)/$m$-$ZrO_2$ before (a) and after hydrogenation (b) are given in Figure 1. In Figure 1a, $ZrO_2$ of the monoclinic phase as well as metallic Ru can be observed for all fresh catalyst samples, indicating that $ZrO_2$ exists as the monoclinic phase and Ru is completely reduced. Notably, when the theoretical molar ratio of Zn to Ru reaches 0.60, characteristic diffraction of ZnO starts to be shown, suggesting that Zn mainly exists as ZnO in Ru-Zn(x)/$m$-$ZrO_2$. This is consistent with that reported by Zhou et al. [9] and Yan et al. [1]. As long as the molar ratio of Zn to Ru is no higher than 0.47, ZnO reflections cannot be detected. This can be rationalized in terms that the amount of ZnO is relatively low, and might not be able to aggregate into a phase. After hydrogenation (Figure 1b) on the

other hand, instead of ZnO, reflections related to $(Zn(OH)_2)_3(ZnSO_4)(H_2O)_3$ can be detected when the molar ratio of Zn to Ru is higher than 0.6. This indicates that ZnO could react with $ZnSO_4$ during the reaction to form the $(Zn(OH)_2)_3(ZnSO_4)(H_2O)_3$ salt. Similarly, no $(Zn(OH)_2)_3(ZnSO_4)(H_2O)_3$ diffraction can be observed when the molar ratio of Zn to Ru is less than 0.47, and the same reason could be applied here.

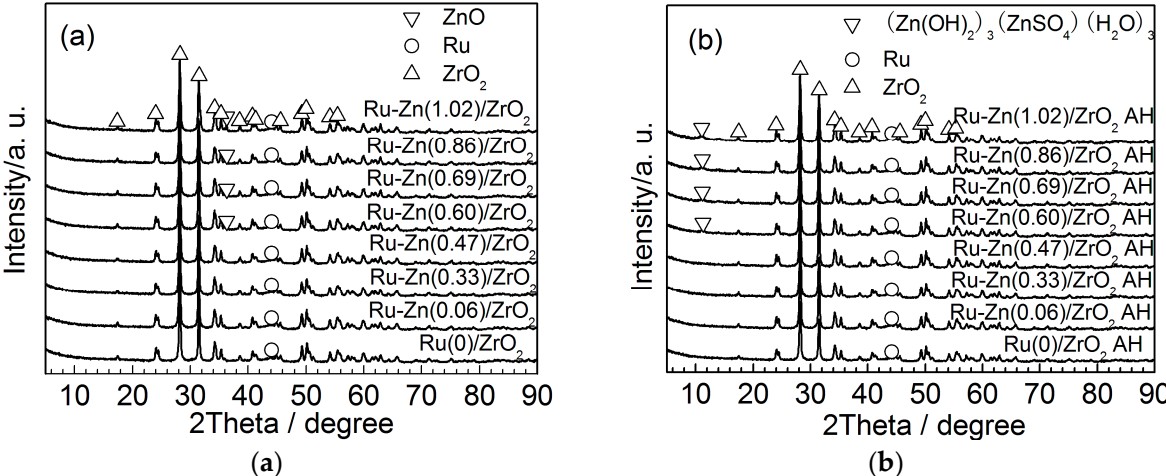

**Figure 1.** XRD patterns of Ru-Zn(x)/ZrO$_2$ before (**a**) and after hydrogenation (**b**).

The composition and texture properties of Ru-Zn($x$)/m-ZrO$_2$ catalysts before and after catalytic experiments, as well as pH values of the slurry after hydrogenation, are listed in Table 1. As can be seen, for the fresh catalysts, the specific surface area, pore diameter, and pore size slightly decreased with increasing Zn content. This implies that some of the macro pores of m-ZrO$_2$ were blocked by the impregnated Ru and ZnO. However, with enhancing the Zn content, pore diameter increased, while the specific surface area and pore volume dropped after the reaction. This might be due to the fact that $(Zn(OH)_2)_3(ZnSO_4)(H_2O)_3$ was generated during the reaction, which covered some of the micro pores of m-ZrO$_2$. On the other hand, it is noticed that the molar ratio of Zn to Ru over Ru-Zn ($x$)/m-ZrO$_2$ after catalytic experiments is higher than that obtained over the fresh catalysts, implying that $Zn^{2+}$ from $ZnSO_4$ or $(Zn(OH)_2)_3(ZnSO_4)(H_2O)_3$ was chemisorbed on the catalyst surface. Additionally, no obvious variation was noticed for $n$ (Zr)/$n$ (Ru), indicating that m-ZrO$_2$ was barely lost during the catalytic experiments. Moreover, the pH value of the slurry after the reaction is less than 6, suggesting that the slurry is acidic. This is mainly attributed to the hydrolysis of $ZnSO_4$. More importantly, the pH value of the slurry after the reaction increased with increasing Zn content, demonstrating that the hydrolysis of $ZnSO_4$ is retarded. This can be rationalized in terms that the formation of $(Zn(OH)_2)_3(ZnSO_4)(H_2O)_3$ led to the decrease in the concentration of $ZnSO_4$.

TEM images and the Ru particle size distribution of Ru-Zn (0.60)/m-ZrO$_2$ catalysts before (a,b) and after (c,d) catalytic experiments are shown in Figure 2. It can be observed from Figure 2a,b that Ru is uniformly dispersed on m-ZrO$_2$, and the particle size of Ru is around 4.5 nm. After the hydrogenation reaction, the particle size of Ru remained at 4.5 nm, indicating that no aggregation of Ru happened during the reaction.

**Table 1.** Composition and texture properties of Ru-Zn(x)/m-ZrO$_2$ catalysts before and after hydrogenation, as well as pH values of the slurry after catalytic experiments.

| Catalyst | $S_{BET}$/(m$^2$·g$^{-1}$) [1] | $V_{pore}$/(cm$^3$·g$^{-1}$) [1] | $d_{pore}$/(nm) [1] | $n_{Zn}/n_{Ru}$ [2] | $n_{Zr}/n_{Ru}$ [2] | pH [3] |
|---|---|---|---|---|---|---|
| ZrO$_2$ | 34 | 0.13 | 16.0 | - | - | - |
| Ru(0)/ZrO$_2$ | 31 | 0.10 | 10.7 | 0 | 5.19 | - |
| Ru-Zn(0.06)/ZrO$_2$ | 30 | 0.09 | 9.8 | 0.06 | 5.30 | - |
| Ru-Zn(0.33)/ZrO$_2$ | 30 | 0.09 | 9.7 | 0.33 | 5.26 | - |
| Ru-Zn(0.47)/ZrO$_2$ | 30 | 0.10 | 9.6 | 0.47 | 5.24 | - |
| Ru-Zn(0.60)/ZrO$_2$ | 30 | 0.09 | 9.7 | 0.60 | 5.38 | - |
| Ru-Zn(0.69)/ZrO$_2$ | 28 | 0.08 | 9.6 | 0.69 | 5.26 | - |
| Ru-Zn(0.86)/ZrO$_2$ | 28 | 0.08 | 9.1 | 0.86 | 5.25 | - |
| Ru-Zn(1.02)/ZrO$_2$ | 29 | 0.08 | 9.1 | 1.02 | 5.16 | - |
| Ru(0)/ZrO$_2$ AH | 30 | 0.11 | 9.5 | 0.20 | 5.21 | 4.25 |
| Ru-Zn(0.06)/ZrO$_2$ AH | 30 | 0.10 | 9.8 | 0.25 | 5.29 | 4.25 |
| Ru-Zn(0.33)/ZrO$_2$ AH | 30 | 0.09 | 9.9 | 0.37 | 5.38 | 5.38 |
| Ru-Zn(0.47)/ZrO$_2$ AH | 30 | 0.08 | 9.8 | 0.55 | 5.36 | 5.38 |
| Ru-Zn(0.60)/ZrO$_2$ AH | 30 | 0.08 | 10.1 | 0.68 | 5.32 | 5.38 |
| Ru-Zn(0.69)/ZrO$_2$ AH | 28 | 0.07 | 10.3 | 0.86 | 5.30 | 5.44 |
| Ru-Zn(0.86)/ZrO$_2$ AH | 29 | 0.08 | 10.6 | 1.10 | 5.26 | 5.82 |
| Ru-Zn(1.02)/ZrO$_2$ AH | 29 | 0.06 | 11.4 | 1.25 | 5.31 | 5.73 |

[1] Determined by N$_2$-sorption; [2] Determined by XRF; [3] Determined by a pH meter at room temperature; AH: after hydrogenation.

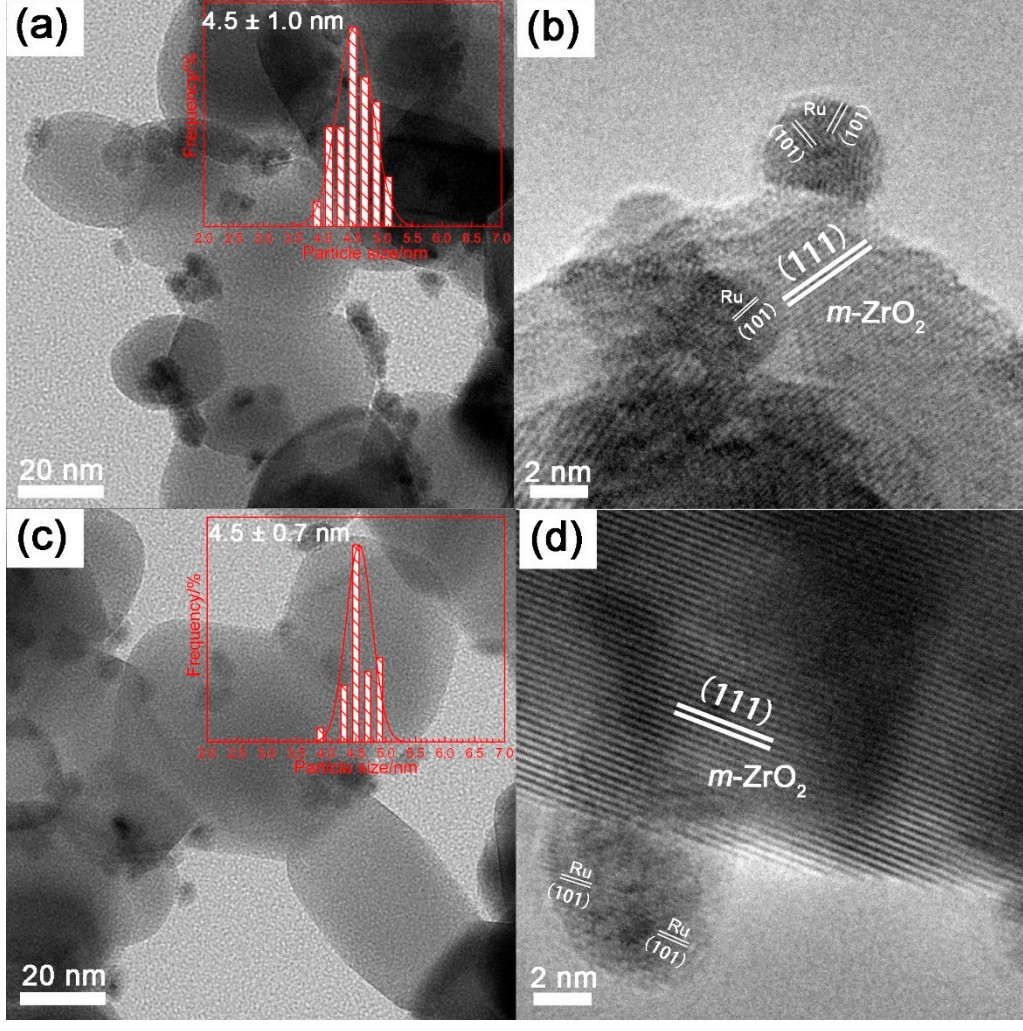

**Figure 2.** TEM images and the Ru particle size distribution of Ru-Zn (0.60)/m-ZrO$_2$ catalysts before (**a**,**b**) and after (**c**,**d**) catalytic experiments.

X-ray photoelectron spectroscopy (XPS) profiles of the Ru-Zn (0.33)/m-ZrO$_2$ catalyst and Ru-Zn (0.60)/m-ZrO$_2$ catalyst after catalytic experiments are presented in Figure 3. It can be observed from Figure 3a that the peak of Ru3d includes Ru3d$_{3/2}$ and Ru3d$_{5/2}$, and the former one is partly overlapped with the peak of C1s. Therefore, Ru3d$_{5/2}$ is selected for further discussion. There are two peaks for Ru3d$_{5/2}$ with binding energy (BE) of 280.3 eV and 281.4 eV, which are attributed to Ru$^0$ and Ru$^{\delta+}$, respectively [26]. The presence of Ru$^{\delta+}$ implies that some Ru lost electrons. Furthermore, it is found that $n$ (Ru$^{\delta+}$)/$n$ (Ru$^0$) over Ru-Zn (0.60)/m-ZrO$_2$ after hydrogenation (AH) is slightly higher than that obtained over Ru-Zn (0.33)/m-ZrO$_2$ AH (i.e., 1.18 vs. 1.09), indicating that more electrons were transferred out of Ru with the enhancement of Zn content. Moreover, as can be seen from Figure 3b, the BE of Zn2p$_{3/2}$ over Ru-Zn (0.33)/m-ZrO$_2$ AH and Ru-Zn (0.60)/m-ZrO$_2$ AH is 1022.0 eV and 1021.1 eV, respectively. However, it is very difficult to justify whether it is metallic Zn or Zn$^{2+}$, since the BEs of metallic Zn and Zn$^{2+}$ for Zn2p$_{3/2}$ are extremely close [27]. Hence, the kinetic energy (KE) of Zn LMM is considered to judge the valence of Zn. It was shown that the KE of Zn LMM for Ru-Zn (0.33)/m-ZrO$_2$ AH and Ru-Zn (0.60)/m-ZrO$_2$ AH is 988.6 eV and 989.2 eV, respectively (Figure 3d). This suggests that the valence of Zn in Ru-Zn/m-ZrO$_2$ AH is positive 2, since the KE of metallic Zn is 992.1 eV [28]. This is in good agreement with the XRD results. In addition, it is noticed that the BE of Zn2p$_{1/2}$ over Ru-Zn (0.60)/m-ZrO$_2$ AH is lower than that observed over Ru-Zn (0.60)/m-ZrO$_2$ AH (i.e., 1021.1 eV vs. 1022.0 eV), but the former's KE is higher than that obtained over the latter. This demonstrates that more electrons were transferred to Zn$^{2+}$ with increasing Zn content. In contrast, the BE of Zr3d$_{5/2}$ over Ru-Zn (0.33)/m-ZrO$_2$ AH and Ru-Zn (0.60)/m-ZrO$_2$ AH is 180.7 eV and 180.6 eV, respectively, which means that the addition of Zn is not able to modify the electronic structure of Zr. However, it is worth mentioning that the BE of Zr3d$_{5/2}$ for ZrO$_2$ observed in this work is clearly lower than that reported in literature (i.e., 182.2 eV) [29], which might be due to the fact that some of the electrons from Ru were transferred to Zr as well. This reveals that there is a strong electronic effect between the active component (Ru) and the support (m-ZrO$_2$).

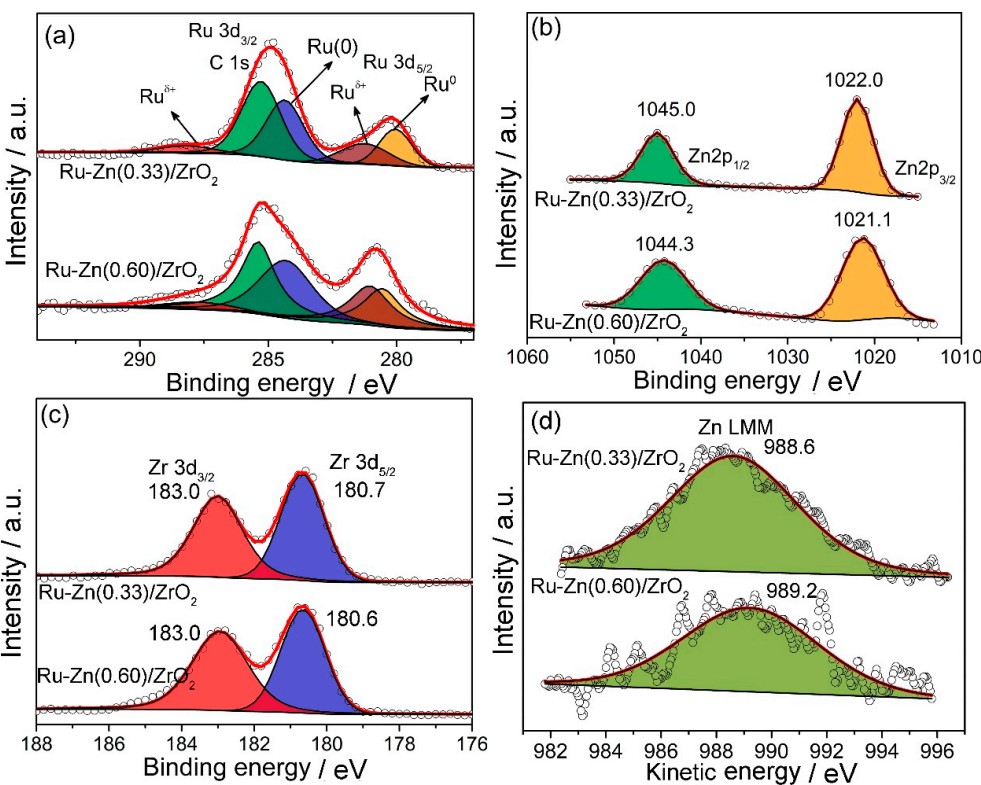

**Figure 3.** X-ray photoelectron spectroscopy (XPS) profiles of Ru-Zn (0.33)/m-ZrO$_2$ catalyst and Ru-Zn (0.60)/m-ZrO$_2$ catalyst after catalytic experiments. (**a**) Ru3d$_{3/2}$ and Ru3d$_{5/2}$; (**b**) Zn2p$_{1/2}$ and Zn2p$_{3/2}$; (**c**) Zr3d$_{3/2}$ and Zr3d$_{5/2}$; (**d**) Zn LMM.

The hydrophilicity of Ru-Zn($x$)/m-ZrO$_2$ after catalytic experiments was further examined, and the corresponding water droplets on each catalyst are illustrated in Figure 4. It is obvious that the water contact angle declined along with raising the Zn content; that is, 139° over Ru/m-ZrO$_2$ AH versus 6° over Ru-Zn (1.02)/m-ZrO$_2$ AH. This indicates that the hydrophilicity of the catalysts is drastically improved by increasing the Zn content. The same assumption can be made from Figure 5, in which fewer catalysts suspended in the organic phase were observed by enhancing the Zn content of the catalyst. The improvement of the wettability of Ru-Zn($x$)/m-ZrO$_2$ catalysts can be rationalized as follows: (1) When ZnO content was increased, more $(Zn(OH)_2)_3(ZnSO_4)(H_2O)_3$ would be generated and chemisorbed on the Ru surface, which directly led to more $Ru^{\delta+}$ being formed during the reaction. Since the lone pair of electrons of oxygen in water molecules are easily linked to the empty d orbitals of Ru, the higher concentration of $Ru^{\delta+}$ resulted in more water molecules being linked to the Ru surface, and thus there was an improvement in the hydrophilicity. (2) The chemisorbed $(Zn(OH)_2)_3(ZnSO_4)(H_2O)_3$ mainly existed as hydrated ions on the Ru surface, which caused the formation of the stagnant water layer. This also leads to the increase in hydrophilicity for the catalysts.

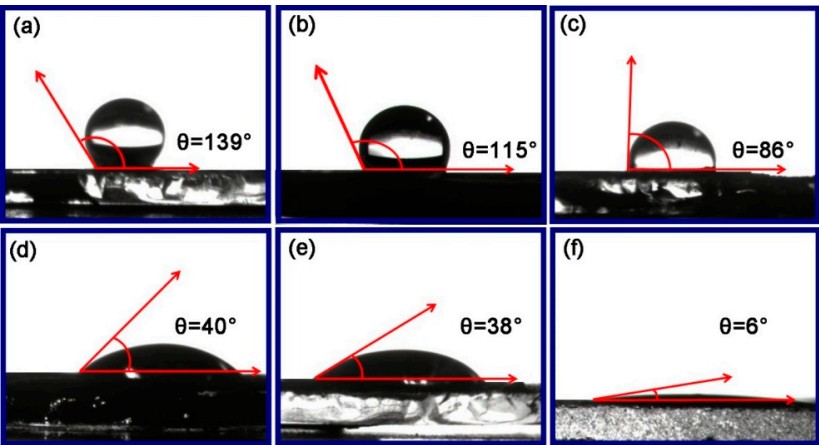

**Figure 4.** Water contact angle of Ru-Zn(x)/m-ZrO$_2$ after catalytic experiments: (**a**) Ru/m-ZrO$_2$ AH; (**b**) Ru-Zn(0.06)/m-ZrO$_2$ AH; (**c**) Ru-Zn(0.33)/m-ZrO$_2$ AH; (**d**) Ru-Zn(0.60)/m-ZrO$_2$ AH; (**e**) Ru-Zn (0.86)/m-ZrO$_2$ AH; (**f**) Ru-Zn (1.02)/m-ZrO$_2$ AH.

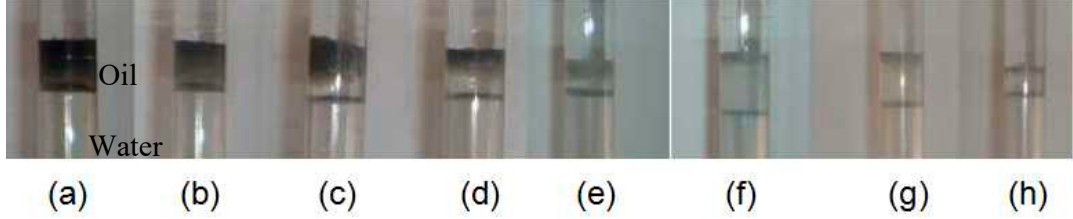

**Figure 5.** The suspend situation of Ru-Zn(x)/m-ZrO$_2$ catalysts in the oil phase after catalytic experiments: (**a**) Ru/m-ZrO$_2$ AH; (**b**) Ru-Zn(0.06)/m-ZrO$_2$ AH; (**c**) Ru-Zn(0.33)/m-ZrO$_2$ AH; (**d**) Ru-Zn (0.47)/m-ZrO$_2$ AH; (**e**) Ru-Zn(0.60)/m-ZrO$_2$ AH; (**f**) Ru-Zn(0.69)/m-ZrO$_2$ AH; (**g**) Ru-Zn (0.86)/m-ZrO$_2$ AH; (**h**) Ru-Zn (1.02)/m-ZrO$_2$ AH.

The catalytic activity and selectivity towards cyclohexene formation over Ru-Zn/m-ZrO$_2$ catalysts with different Zn content is illustrated in Figure 6. It can be clearly seen that the catalytic activity towards benzene conversion over Ru-Zn/m-ZrO$_2$ catalysts was inhibited by increasing the Zn content; that is, when the molar ratio of Zn to Ru grew from 0 to 1.02, catalytic activity towards benzene conversion dropped from 99.8% to 14.9% in 20 min of reaction time. On the contrary, catalytic selectivity towards cyclohexene formation increased along with Zn content. Notably, when the molar ratio of Zn to Ru exceeded 0.6, catalytic activity towards benzene conversion was drastically suppressed (e.g., 88.8% of benzene conversion over Ru-Zn (0.60)/m-ZrO$_2$ vs. 21.7% of benzene conversion over

Ru-Zn (1.02)/m-ZrO$_2$ after 40 min of catalytic experiments), while no significant improvement for the selectivity towards cyclohexene synthesis was observed (e.g., 68.0% of cyclohexene selectivity over Ru-Zn (0.60)/m-ZrO$_2$ vs. 88.8% of benzene conversion over Ru-Zn (1.02)/m-ZrO$_2$ after 40 min of catalytic experiments). When $n$(Zn)/$n$(Ru) is 0.6, a 60.9% cyclohexene yield was achieved within 35 min, which is the highest yield of cyclohexene ever reported over Ru-Zn/ZrO$_2$ [1,9,25]. Combined with the characterization results, the effect of Zn content can be concluded as follows: If the surface of Ru is hydrophobic, the catalyst mainly stays in the oil phase, in which the adsorbed benzene tends to be completely hydrogenated into cyclohexane, since desorption of the formed cyclohexene hardly proceeds in the oil phase. Therefore, When Zn content increases, more (Zn(OH)$_2$)$_3$(ZnSO$_4$)(H$_2$O)$_3$ would be generated and chemisorbed on the Ru surface, improving the wettability of the catalyst. The result of this is that the hydrophilic catalyst easily moves from the oil phase into the aqueous phase and stays there. It is well known that the solubility of cyclohexene in water is weaker than that of benzene [30], thus the synthesized cyclohexene in aqueous phase would be transferred into the oil phase spontaneously. The formed cyclohexene is difficult to be re-adsorbed on the hydrophilic Ru surface, leading to the inhibition of its further hydrogenation, and thus improving the selectivity towards cyclohexene formation. Besides, when the hydrogenation of benzene takes place in the aqueous phase, the generated cyclohexene can be stabilized by the hydrogen bond formed between cyclohexene and water molecules [31]. As demonstrated in Figure 7, two types of hydrogen bond between cyclohexene and water molecules could be formed in the aqueous phase. However, as the hydrophilicity further increases with increasing n (Zn)/n (Ru), there is less retention time that Ru-Zn/ZrO$_2$ spends in the oil phase. This leads to a fall in benzene adsorption of Ru-Zn/ZrO$_2$ catalysts, and catalytic activity towards benzene conversion declines (Figure 6a). Therefore, there is an optimum molar ratio of Zn to Ru for Ru-Zn/ZrO$_2$ catalysts; that is, n (Zn)/n (Ru) = 0.6 (Figure 7, "catalyst B"). When Zn content is very insufficient, as in "catalyst A" in Figure 7, the generated cyclohexene is likely hydrogenated into cyclohexane, leading to a relatively low selectivity towards cyclohexene. In contrast, when Zn content is excessive (Figure 7 "catalyst C"), adsorption of benzene becomes quite challenging, which causes a drastic decrease of catalytic activity towards benzene conversion.

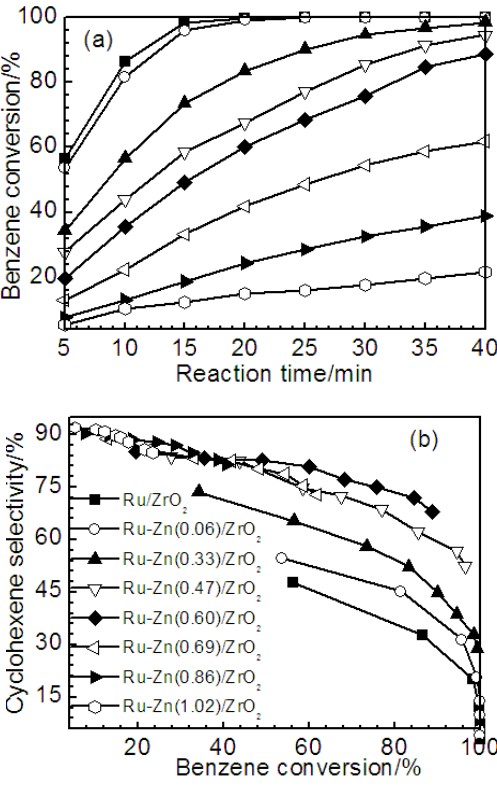

**Figure 6.** *Cont.*

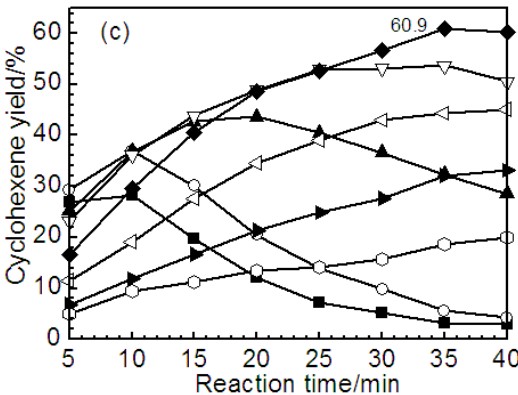

**Figure 6.** Catalytic activity towards selective hydrogenation of benzene over Ru-Zn/m-ZrO$_2$ catalysts with different Zn content ($m_{cat}$ = 1.2 g, $m_{ZnSO4}$ = 50.0 g, $v_{H2O}$ = 280 cm$^3$, $v_{benzene}$ = 140 cm$^3$, T = 423 K, $p_{H2}$ = 5.0 MPa). (**a**) Benzene conversion as function of reaction time; (**b**) Cyclohexene selectivity as function of benzene conversion; (**c**) Cyclohexene yield as function of reaction time.

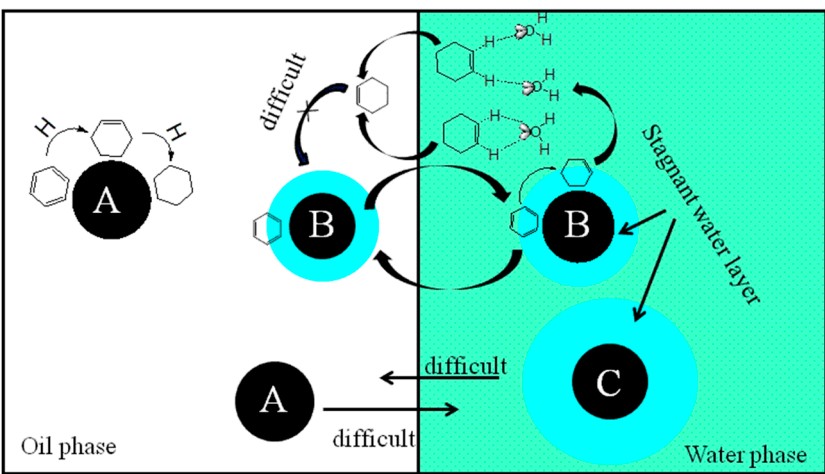

**Figure 7.** Two types of hydrogen bond formed between cyclohexene and water molecules, as well as the hydrogenation scheme both in the oil phase and aqueous phase, over Ru-Zn/ZrO$_2$ catalysts with different wettability.

## 2.2. Effect of ZrO$_2$ as Support and Dispersant

Figure 8 demonstrates the XRD patterns of the supported Ru-Zn (0.60)/m-ZrO$_2$ catalyst and the unsupported Ru-Zn (0.60) catalyst before (a) and after (b) catalytic experiments. As with m-ZrO$_2$ supported samples, metallic Ru and ZnO reflections were observed for the unsupported Ru-Zn catalyst, which again proves that the valence of Ru and Zn is 0 and positive 2, respectively. Moreover, the characteristic diffraction of (Zn(OH)$_2$)$_3$(ZnSO$_4$)(H$_2$O)$_3$ was also pronounced over the unsupported sample, indicating that ZrO$_2$ as the support plays no role in the formation of (Zn(OH)$_2$)$_3$(ZnSO$_4$)(H$_2$O)$_3$.

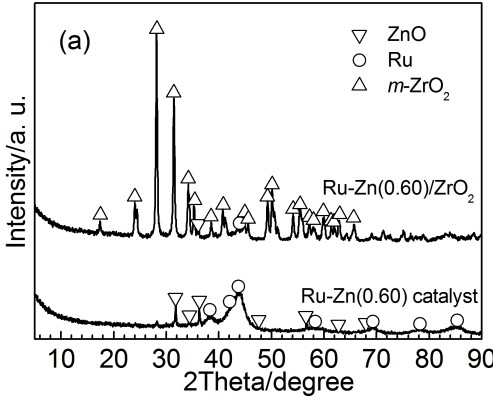 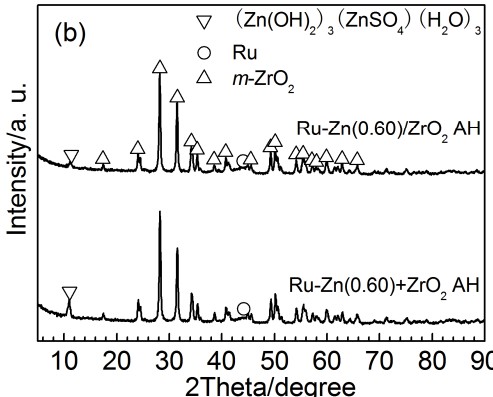

**Figure 8.** XRD patterns of the supported Ru-Zn(0.60)/m-ZrO$_2$ catalyst and the unsupported Ru-Zn (0.60) catalyst before (**a**) and after (**b**) catalytic experiments.

The texture properties and composition of the supported Ru-Zn (0.60)/m-ZrO$_2$ catalyst and the unsupported Ru-Zn (0.60), as well as the pH values of the slurry, after catalytic experiments are given in Table 2. It is found that unsupported Ru-Zn (0.60) catalyst demonstrated similar texture properties to that obtained over parent m-ZrO$_2$ after the catalytic experiment. This can be attributed to the fact that the added m-ZrO$_2$ was 10 times the amount of catalyst used and was mechanically mixed with the catalyst during the reaction. Moreover, an increase in the molar ratio of Zn to Ru was also shown for the unsupported Ru-Zn catalyst after catalytic experiments, indicating that (Zn(OH)$_2$)$_3$(ZnSO$_4$)(H$_2$O)$_3$ was formed as well. Interestingly, it was noticed that n (Zn)/n (Ru) over Ru-Zn (0.60)/ZrO$_2$ is slightly higher than that obtained over the unsupported sample after the reaction; that is, 0.68 versus 0.65. This implies that m-ZrO$_2$ as the support benefits the adsorption of (Zn(OH)$_2$)$_3$(ZnSO$_4$)(H$_2$O)$_3$ on the Ru surface. Furthermore, the pH value of the slurry with application of Ru-Zn (0.60)/ZrO$_2$ is slightly higher than that observed with unsupported Ru-Zn (0.60), suggesting that more Zn$^{2+}$ was chemisorbed on the catalyst surface as well.

**Table 2.** Texture properties and composition of the supported Ru-Zn(0.60)/ZrO$_2$ catalyst and the unsupported Ru-Zn (0.60), as well as the pH values of the slurry, after hydrogenation.

| Catalyst | $S_{BET}$/(m$^2$·g$^{-1}$) [1] | $V_{pore}$/(cm$^3$·g$^{-1}$) [1] | $d_{pore}$/(nm) [1] | $n_{Zn}/n_{Ru}$ [2] | $n_{Zr}/n_{Ru}$ [2] | pH [3] |
|---|---|---|---|---|---|---|
| Ru-Zn(0.60)/ZrO$_2$ | 30 | 0.09 | 9.7 | 0.60 | 5.38 | - |
| Ru-Zn(0.60)/ZrO$_2$ AH | 30 | 0.08 | 10.1 | 0.68 | 5.33 | 5.38 |
| Ru-Zn(0.60) | 65 | 0.19 | 11.7 | 0.61 | 0 | - |
| Ru-Zn(0.60)+ZrO$_2$ AH | 33 | 0.12 | 14.6 | 0.65 | 5.35 | 5.29 |
| ZrO$_2$ | 34 | 0.13 | 16.0 | - | - | - |

[1] Determined by N$_2$-sorption; [2] Determined by XRF; [3] Determined by a pH meter at room temperature; AH: after hydrogenation.

Figure 9 shows the TEM images and Ru particle size distribution of the unsupported Ru-Zn (0.60) catalyst before and after the reaction. As can be observed from Figure 9a,b, fresh Ru-Zn (0.60) displays a circular or elliptical shape, of which the particle size is around 4.5 nm. In addition, analogous to that observed over the supported sample, no obvious change in the particle size of Ru-Zn (0.6) after the catalytic experiment is demonstrated (Figure 9c). However, unlike Ru-Zn (0.60)/m-ZrO$_2$, it is noticed that Ru-Zn (0.6) is clearly not uniformly dispersed by adding m-ZrO$_2$ as the dispersant (Figure 9d). This suggests that in comparison with using m-ZrO$_2$ as the dispersant, m-ZrO$_2$ could contribute better to the dispersion of Ru when it is utilized as the support.

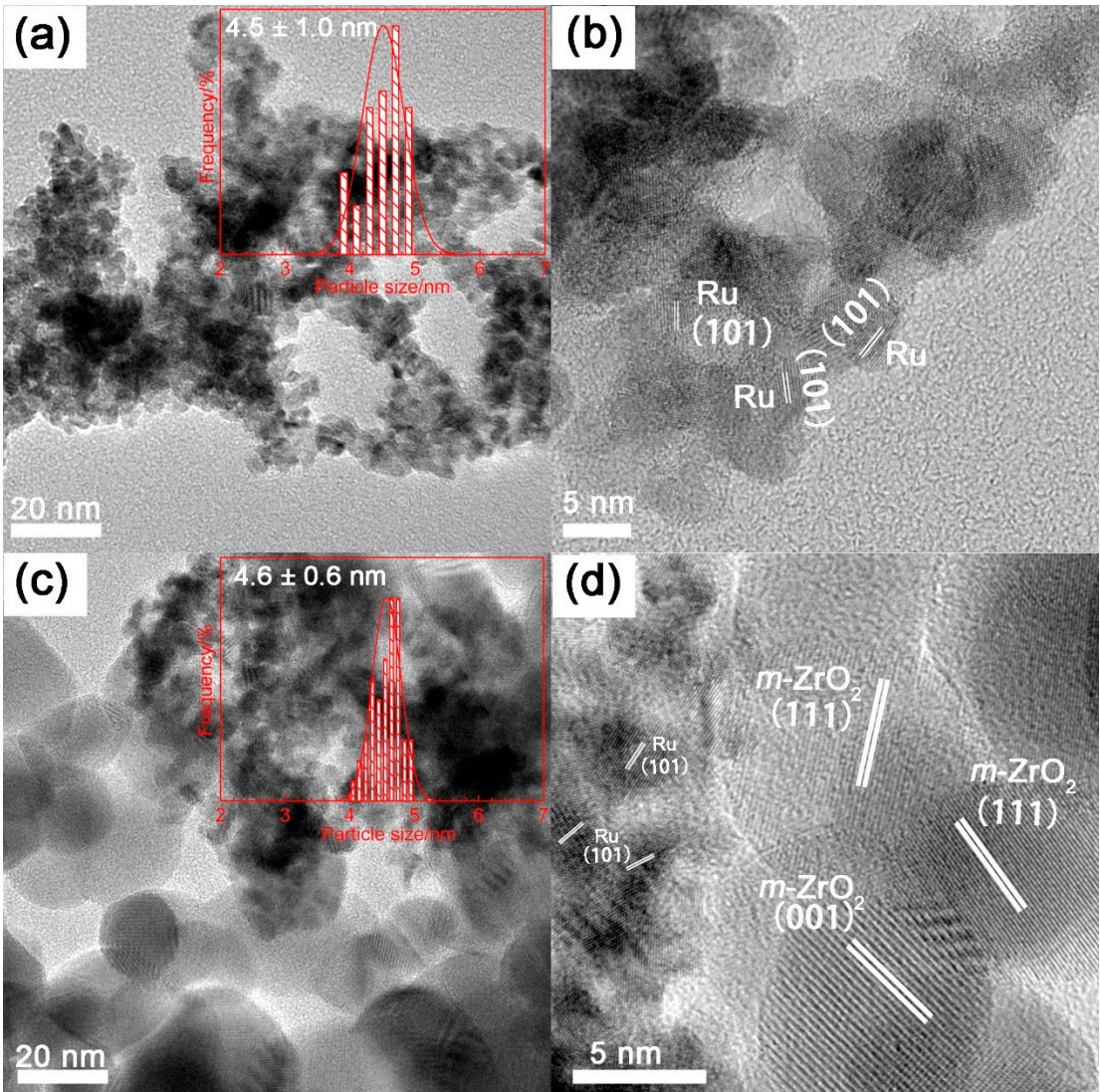

**Figure 9.** TEM images and Ru particle size distributions of the unsupported Ru-Zn (0.60) catalyst with m-ZrO$_2$ as the dispersant before (**a**,**b**) and after (**c**,**d**) catalytic experiments.

XPS profiles of the unsupported Ru-Zn (0.60) catalyst with m-ZrO$_2$ as the dispersant after catalytic experiments are demonstrated in Figure 10. As can be observed from Figure 10a, two peaks related to the BE of Ru3d$_{5/2}$ were observed at 280.3 eV and 281.4 eV, which are attributed to Ru$^0$ and Ru$^{\delta+}$, respectively. This is consistent with that obtained over the corresponding supported sample (Ru-Zn (0.60)/m-ZrO$_2$). Moreover, from Figure 10b, the BE of Zn2p$_{3/2}$ over unsupported Ru-Zn (0.60) is observed to be 1021.5 eV, which is slightly higher than that demonstrated over Ru-Zn (0.60)/m-ZrO$_2$ (i.e., 1021.1 eV). Additionally, the KE of Zn LMM over unsupported Ru-Zn (0.60) is slightly lower than that observed over Ru-Zn (0.60)/m-ZrO$_2$; that is, 989.0 eV versus 989.2 eV. This can be rationalized in terms that more (Zn(OH)$_2$)$_3$(ZnSO$_4$)(H$_2$O)$_3$ was chemisorbed on Ru-Zn (0.60)/m-ZrO$_2$ than that on unsupported Ru-Zn (0.60), thus leading to more electrons being transferred from Ru to Zn. This is in a good agreement with the XRF results. Additionally, the BE of Zr3d$_{5/2}$ over unsupported Ru-Zn (0.60) with m-ZrO$_2$ as the dispersant was normalized to be 182.0 eV, which is close to that reported in literature (i.e., 182.2 eV) [29]. This suggests that there is no obvious electronic effect between the active component (Ru) and m-ZrO$_2$ as the dispersant. It is therefore deemed that no electrons were transferred from Ru to Zr by mechanical mixing of Ru-Zn (0.60) with m-ZrO$_2$.

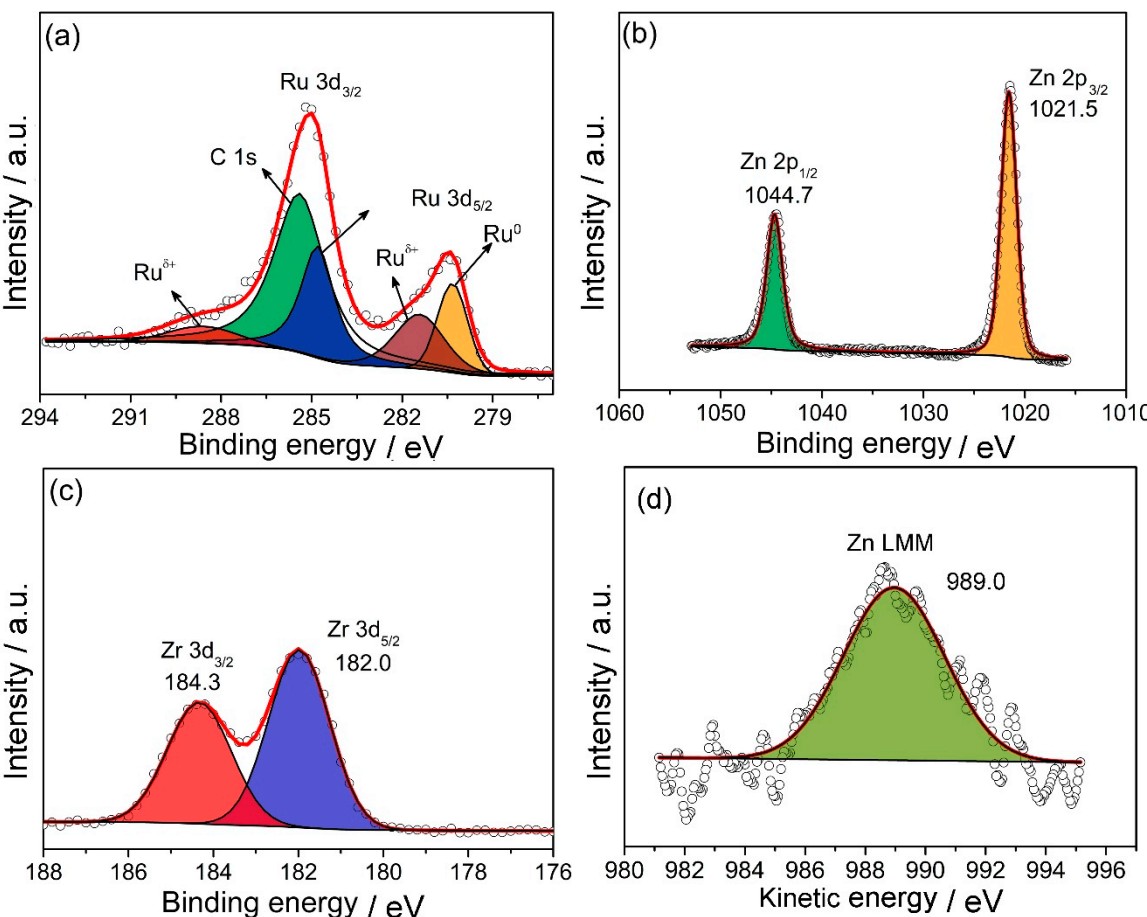

**Figure 10.** XPS profiles of the unsupported Ru-Zn (0.60) catalyst with m-ZrO$_2$ as the dispersant after catalytic experiments. (**a**) Ru3d$_{3/2}$ and Ru3d$_{5/2}$; (**b**) Zn2p$_{1/2}$ and Zn2p$_{3/2}$; (**c**) Zr3d$_{3/2}$ and Zr3d$_{5/2}$; (**d**) Zn LMM.

The H$_2$-TPR profiles of Ru-Zn (0.60)/m-ZrO$_2$ and unsupported Ru-Zn (0.60) after catalytic experiments are given in Figure 11. The reduction peak of Ru over Ru-Zn (0.60)/m-ZrO$_2$ can be observed at around 360 K, which is higher than that shown over unsupported Ru-Zn (0.60) (i.e., 344 K). This indicates that the interaction between Ru and the support (m-ZrO$_2$) is stronger than that between Ru and the dispersant. This is in agreement with the XPS results.

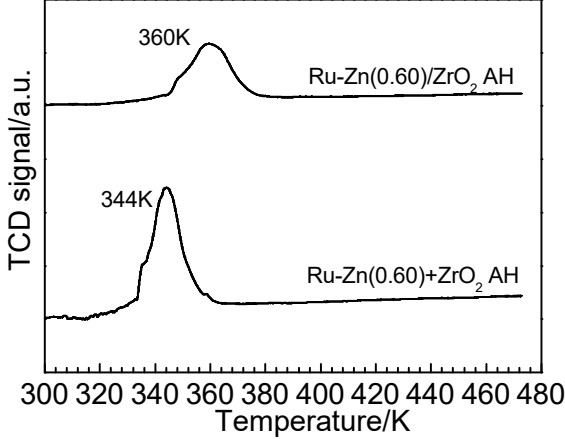

**Figure 11.** H$_2$-temperature-programmed reduction profiles of Ru-Zn(0.60)/m-ZrO$_2$ and unsupported Ru-Zn (0.60) after catalytic experiments.

The wettability of unsupported Ru-Zn(0.60) after catalytic experiments was also examined, and the water droplet on the catalyst is illustrated in Figure 12. As presented, the contacted angle over unsupported Ru-Zn(0.60) is 52°, which is higher than that observed over Ru-Zn(0.60)/ZrO$_2$ (i.e., 40°). This demonstrates that hydrophilicity of Ru-Zn with m-ZrO$_2$ as the dispersant is weaker than that with m-ZrO$_2$ as the support. This is mainly due to the fact that no electron was transferred from Ru to Zr, causing less Ru$^{\delta+}$ to be generated. This leads to the fact that less water molecules are linked to the Ru surface, thus decreasing the hydrophilicity of the catalyst surface.

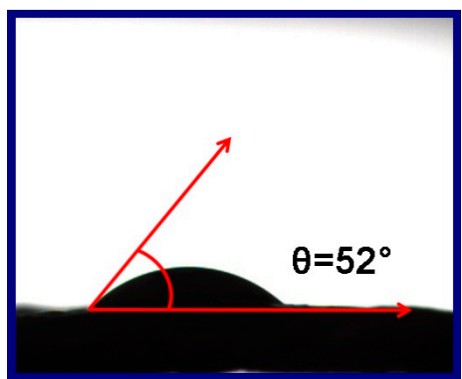

**Figure 12.** Water contact angle of unsupported Ru-Zn (0.60) after catalytic experiments.

Catalytic activity and selectivity towards cyclohexene formation over unsupported Ru-Zn (0.60), as well as Ru-Zn (0.60)/m-ZrO$_2$, are presented in Figure 13. As can be seen, both catalytic activity towards benzene conversion, and selectivity towards cyclohexene, over unsupported Ru-Zn (0.60), are lower than that achieved over Ru-Zn (0.60)/m-ZrO$_2$; that is, only 69.5% of benzene conversion and 71.5% of cyclohexene selectivity were obtained over unsupported Ru-Zn (0.60), while Ru-Zn (0.60)/m-ZrO$_2$ gave 84.6% of benzene conversion and 71.5% of selectivity to cyclohexene after 35 min of reaction time. This can be attributed to two main reasons: (1) Ru can be highly dispersed on m-ZrO$_2$ when m-ZrO$_2$ is utilized as the support, while the dispersion of Ru is less uniform when m-ZrO$_2$ is applied as the dispersant. This results in more active Ru sites being available during the reaction, and further improves the catalytic activity towards benzene conversion. (2) m-ZrO$_2$ supported Ru-Zn (0.60) benefits the adsorption of (Zn(OH)$_2$)$_3$(ZnSO$_4$)(H$_2$O)$_3$, enhancing the hydrophilicity of the catalyst surface. Therefore, more Ru-Zn (0.60)/m-ZrO$_2$ would stay in the aqueous phase than that happens for the unsupported sample, which helps the desorption of the generated cyclohexene and inhibits its further hydrogenation. Hence, the highest cyclohexene yield of 60.9% was achieved over Ru-Zn (0.60)/m-ZrO$_2$, while only 50% of the maximum cyclohexene yield was obtained over unsupported Ru-Zn (0.60).

The reusability of Ru-Zn (0.60)/m-ZrO$_2$ was investigated under the same reaction conditions without further regeneration (Figure 14). It can be observed that benzene conversion as well as cyclohexene selectivity is maintained above 80% and 70%, respectively, after 17 iterations of the catalytic experiments. Thus, the catalytic system over Ru-Zn (0.60)/m-ZrO$_2$ shows a good reusability, indicating that this catalyst possesses great potential for industrial application in selective hydrogenation of benzene towards cyclohexene production.

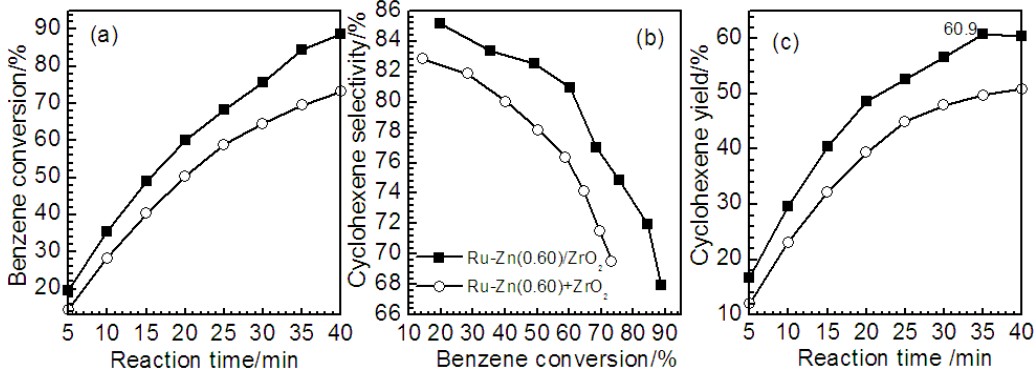

**Figure 13.** Catalytic activity towards selective hydrogenation of benzene over unsupported Ru-Zn (0.60) as well as Ru-Zn(0.60)/m-ZrO$_2$ (m(Ru-Zn (0.60)/m-ZrO$_2$) = 1.2 g, or m(Ru-Zn (0.60)) = 0.2 g and m(ZrO$_2$) = 1.0 g; m$_{ZnSO4}$ = 50.0 g; v$_{H2O}$ = 280 cm$^3$; v$_{benzene}$ = 140 cm$^3$; T = 423 K; p$_{H2}$ = 5.0 MPa). (**a**) Benzene conversion as function of reaction time; (**b**) Cyclohexene selectivity as function of benzene conversion; (**c**) Cyclohexene yield as function of reaction time.

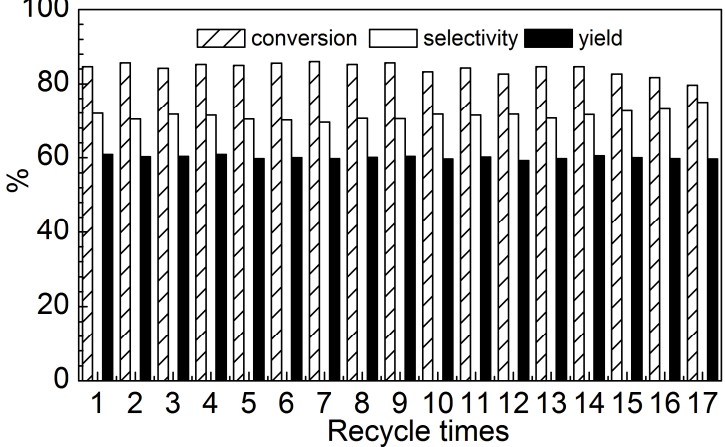

**Figure 14.** Reusability of Ru-Zn (0.60)/m-ZrO$_2$ for selective hydrogenation of benzene, including benzene conversion, cyclohexene selectivity, and yield (v$_{H2O}$ = 280 cm$^3$, v$_{benzene}$ = 140 cm$^3$, T = 423 K, p$_{H2}$ = 5.0 MPa, t$_{reaction}$ = 40 min).

## 3. Materials and Methods

### 3.1. Chemicals

All chemicals were directly used without any further purification. RuCl$_3$·3H$_2$O was delivered from Sino-Platinum Co., Ltd. (Kunming, China). ZnSO$_4$·7H$_2$O was purchased from the Fuchen Chemical Reagent Factory (Tianjin, China). NaOH and benzene were commercially obtained from the Kemiou Chemical Reagent Co., Ltd. (Tianjin, China). m-ZrO$_2$ was synthesized according to the literature [32]. Distilled water was applied in all experiments.

### 3.2. Preparation of Catalysts

Ru-Zn/m-ZrO$_2$ catalysts were synthesized as follows. First, 0.45 g of RuCl$_3$·3H$_2$O and 0.28 g of ZnSO$_4$·7H$_2$O were dissolved in 1 cm$^3$ of deionized water. With continuous stirring, the aqueous solution was added dropwise onto 1.0 g of m-ZrO$_2$ powder. Subsequently, the solid was dried at 323 K for 3 h, and then calcined at 423 K in a Muffle furnace for another 3 h. After that, the calcined sample, together with 200 cm$^3$ of 5 wt. % NaOH aqueous solution, was transferred into a 1000 cm$^3$ Hastelloy autoclave at 423 K under 5.0 MPa of hydrogen and a stirring speed of 800 min$^{-1}$ for 1 h. Ru(OH)$_3$ was generated when the NaOH was mixed with the calcined sample in the autoclave during the

temperature rising procedure. Then the black powder was cooled down to room temperature, washed with distilled water until neutral, and vacuum-dried. Subsequently, 1.2 g of fresh Ru-Zn/m-ZrO$_2$ catalyst was obtained with a theoretical molar ratio of Zn to Ru of 0.06, and was denoted as Ru-Zn (0.60)/m-ZrO$_2$. Additionally, Ru-Zn (*x*)/m-ZrO$_2$ catalysts were synthesized via the same procedure by modifying the usage of ZnSO$_4$·7H$_2$O, where *x* refers to the theoretical molar ratio of Zn to Ru. The preparation procedure of Ru-Zn (*x*)/m-ZrO$_2$ catalysts is illustrated in Scheme 1.

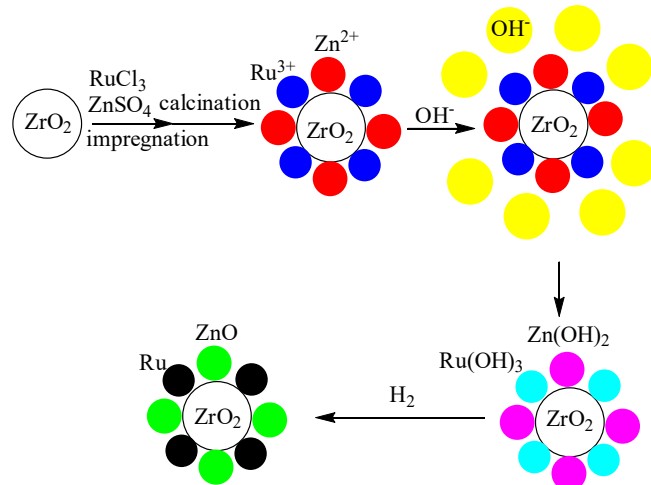

**Scheme 1.** Preparation procedure of Ru-Zn (x)/m-ZrO$_2$ catalysts.

Unsupported Ru-Zn catalyst was prepared using a reported co-precipitation method [8]. First, 0.45 g of RuCl$_3$·3H$_2$O precursor and 0.28 g of ZnSO$_4$·7H$_2$O were dissolved in 100 cm$^3$ of distilled water. Then 100 cm$^3$ of NaOH (5 wt.%) aqueous solution was added at 353 K with continuous stirring for 2 h, followed by moving all solids and solutions into a 1000 mL Hastelloy autoclave for a reduction procedure at 423 K with 5.0 MPa of hydrogen and a stirring speed of 800 rpm for 3 h. After reduction, the sample was cooled, and the fresh catalyst was gained by washing with deionized water to neutral and vacuum-drying. The prepared Ru-Zn catalysts were denoted as Ru-Zn (0.60), where 0.60 stands for the theoretical molar ratio of Zn to Ru.

### 3.3. Catalytic Experimental Procedure

All hydrogenation reactions took place in a 1000 mL GS-1 type Hastelloy autoclave. In a typical hydrogenation reaction, 1.2 g Ru-Zn (*x*)/m-ZrO$_2$ catalyst, 50.0 g ZnSO$_4$·7H$_2$O (0.62 mol L$^{-1}$), as well as 280 cm$^3$ distilled water, were added in the reactor. Then the autoclave was charged using N$_2$ to 4 MPa 4 times, which was followed by purification with H$_2$ to 4 MPa for another 4 times. Then, the reactor was heated to 423 K under 5.0 MPa of H$_2$ with a stirring speed of 800 min$^{-1}$, followed by pouring 140 cm$^3$ of benzene and modifying the stirring speed to 1400 rpm (to get rid of the mass transfer limitation) [33]. After that, each liquid sample was taken from the reactor every 5 min. All withdrawn samples were analyzed using GC-FID from the Hangzhou Kexiao Chemical Instrument and Equipment Co., Ltd. (Hangzhou, China). As with the evaluation of the unsupported Ru-Zn catalyst, 0.2 g of catalyst sample as well as 1 g of m-ZrO$_2$ were individually added instead of 1.2 g of Ru-Zn (*x*)/m-ZrO$_2$ catalyst, and the rest of the procedure was the same. The catalytic activity towards benzene conversion and selectivity towards cyclohexene were calculated using the calibration area normalization method, and the correlation coefficient (R$^2$) of all compounds was higher than 0.99. After each reaction, the organic phase was removed via a separating funnel, and other parts were recharged into the autoclave to investigate the reusability of the catalysts through identical experimental procedures. Statistical validation was evaluated by conducting catalytic experiments over Ru-Zn (0.60)/m-ZrO$_2$ in three separate runs under the same reaction conditions, and the standard deviation for cyclohexene yield was

calculated at 1.3% after 35 min of reaction time. For the used catalysts that needed to be characterized, samples were filtered and washed until the filtrate became neutral and no $Zn^{2+}$ could be detected. Then solid samples were dried in Ar flow at 373 K and stored in ethanol, ready for further characterization.

### 3.4. Catalysts Characterization

X-ray diffraction (XRD) patterns for the fresh and spent catalysts were measured using an X'Pert Pro instrument from Philips (Almelo, The Netherland) at room temperature. The diffracted intensity of Cu-K$\alpha$ radiation ($\lambda$ = 0.154 nm) was recorded in the range of 2$\theta$ from 5° to 90°, with a step size of 0.03°. In addition, textural properties were analyzed using the Nova 1000 e-Physisorption Analyzer (Quantachrome Instruments, Boynton Beach, FL, USA). Before measurements, all the samples were evacuated at 523 K under vacuum pressure for 2 h, then the isotherms were taken at 77 K. The specific surface area ($S_{BET}$) was determined using the Brunauer-Emmett-Teller (BET) model. Furthermore, elemental analysis was conducted via X Ray Fluorescence (XRF) using a S4 Pioneer instrument (Bruker AXS, Karlsruhe, Germany). Additionally, X-ray photoelectron spectroscopy (XPS) using a PHI Quantera SXM instrument from Ulvac-Phi (Kangawa, Japan) was utilized for analyzing the valence state of Ru and Zn on the catalyst surface. Al K$\alpha$ (Eb = 1486.6 eV) was selected as the source of radiation and the vacuum degree was set to 6.7 $\times$ 10$^{-8}$ Pa. The C1s (Eb = 284.8 eV) line as the binding energy reference was used for calibrating and correcting the energy scale. Furthermore, JEOL JEM 2100 transmission electron microscopy (TEM) combined with an energy dispersive spectrometer (EDS) (Akishima, Tokyo, Japan) was applied to investigate the dispersion of the catalysts, as well as the particle size. To investigate the hydrophilicity of the catalyst surface, a contact angle meter (JC2000 C1, Powereach, Shanghai, China) was used to measure water contact angle values (CAs) at ambient temperature for each sample. Moreover, temperature programmed reduction (TPR) was conducted with an Autosorb-IQ from Quantachrome (Boynton Beach, FL, USA). Typically, prior to reduction, a 10 mg sample was oxidized in flowing synthetic air (flow rate: 30 cm$^3$ min$^{-1}$) while being heated to 423 K and held for 1 h. After cooling in an Argon stream (flow rate: 30.0 cm$^3$ min$^{-1}$) to 293 K, the sample was treated for another 2 h. Then an Ar stream containing 10 Vol % $H_2$ was introduced instead (30 cm$^3$ min$^{-1}$), while being heated to 573 K (10 K min$^{-1}$) and held for 1 h. The hydrogen consumption was recorded and determined using a standard CuO calibration.

## 4. Conclusions

m-$ZrO_2$ supported Ru-Zn catalysts, as well as unsupported Ru-Zn catalyst with m-$ZrO_2$ as the dispersant, were evaluated for selective hydrogenation of benzene towards cyclohexene formation. Zn mainly exists as ZnO in Ru-Zn catalysts. Moreover, by increasing the Zn content in the Ru-Zn/m-$ZrO_2$ catalyst, more $(Zn(OH)_2)_3(ZnSO_4)(H_2O)_3$ was generated and chemisorbed on the surface of the catalysts. This decreases the catalytic activity towards benzene (99.8% over Ru/m-$ZrO_2$ vs. 14.9% over Ru-Zn (1.02)/m-$ZrO_2$ after 20 min of reaction) and improves the selectivity to cyclohexene formation (68.0% over Ru-Zn (0.06)/m-$ZrO_2$ vs. 88.8% over Ru-Zn (1.02)/m-$ZrO_2$ after 40 min of reaction). When the molar ratio of Zn to Ru reached 0.6, the highest cyclohexene yield of 60.9% was achieved. In addition, when m-$ZrO_2$ was applied as the dispersant instead of being utilized as the support, both catalytic activity towards benzene conversion and selectivity to cyclohexene were suppressed; that is, only 69.5% of benzene conversion and 71.5% of cyclohexene selectivity were obtained over unsupported Ru-Zn (0.60), while Ru-Zn (0.60)/m-$ZrO_2$ gave 84.6% of benzene conversion and 71.5% of selectivity to cyclohexene after 35 min of reaction time. This can be rationalized in terms that m-$ZrO_2$ as the support has a strong interaction with the Ru-Zn catalyst, benefiting the dispersion of Ru and the chemisorption of $(Zn(OH)_2)_3(ZnSO_4)(H_2O)_3$. Notably, the reusability of Ru-Zn (0.60)/m-$ZrO_2$ was evaluated for selective hydrogenation of benzene, and no obvious decrease in catalytic activity towards cyclohexene formation was observed after 17 reaction iterations, over which at least a 59.3% cyclohexene yield can be achieved.

**Author Contributions:** H.S., Z.C., and Z.P. conceived and designed the experiments; H.L. performed the experiments; Z.L. and S.L. analyzed the data; L.C. contributed reagents, materials, and analysis tools; H.S. wrote the paper.

**Funding:** This research received no external funding.

**Acknowledgments:** This work was supported by the National Nature Science Foundation of China (21273205), the Key Scientific Research Project of Henan Province (18A180018), the Environmental Catalysis Innovative Research Team of Zhengzhou Normal University (702010), and the Student Innovation Program of Zhengzhou Normal University (DCZ2017014).

**Conflicts of Interest:** The authors declare no conflict of interest.

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
