# Peer review of "Selective Hydrogenation of Benzene to Cyclohexene over Ru-Zn Catalysts: Investigations on the Effect of Zn Content and ZrO2 as the Support and Dispersant"

_catalysts, doi:10.3390/catal8110513_

Reviewer 1 Report

This is very interesting paper dealing with a topic which has both fundamental research needs and eventual technology transfer (improved catalytic formulations).

The literature review is fairly exhaustive and the justification is well explained and sound.

The authors utilize appropriately the analytical techniques and the observations are successfully combine to provide scientifically sound explanations.

Statistical validation is not provided with and this is probably the only weakness I see. For example,the authors should reproduce a number of the experiments and report deviations. The order of these deviations could be used to validate that the observed differences are statistically significant. 

Overall, I believe that this is a publishable manuscript if the authors bring forward this addition.

Author Response

Thank you very much for your kind suggestion. Statistical validation is evaluated by conducting catalytic experiments over Ru-Zn (0.60)/m-ZrO2 in three separate runs under the same reaction conditions, and the standard deviation for cyclohexene yield is given in the experimental section 3.3.

Reviewer 2 Report

The manuscript comprises some interesting aspects of Ru-Zn catalysts for the selective hydrogenation of benzene to cyclohexene

 Some topics should be added, corrected or clarified:

 Abstract:

- Please specify the employed zirconia and explain the term “m-ZrO2”.

- “… transition electron microscopy (TEM) …”

 Introduction:

Please write uniformly “Zn” or “zinc” throughout the manuscript.

 Chapter 2.1.:

- p. 3, l. 102: “… Table 1 …”

- p. 4, l. 116: “… the decrease of …”

- Fig. 3: The particle size distribution is hardly visible.

- Fig.s 4 and 11: “Kinetic energy …”

- p. 7, l. 190: “… higher than 0.6, catalytic benzene conversion …”

 Chapter 2.2.:

- p. 9, l. 243: “… Table 2. …”

- Fig. 10: Please check the figure caption. It should be mentioned that ZrO2 is also part of the catalyst system.

- p. 12, l. 322: “… highest cyclohexene yield of …”

 Chapter 3.2:

Scheme 1: Please explain the step after NaOH addition briefly.

 Chapter 3.3:

- What is the role of ZnSO4? Please explain this anywhere in the manuscript.

- p. 14, l. 383: “… until the filtrate became neutral and …”

 Author Response

Thank you very much for your helpful comments. Please find the response as followings:

 Point 1: Please specify the employed zirconia and explain the term “m-ZrO2”.

 Response 1: They are added in the abstract as “i.e., employing as an additive during the reaction,” and “(monoclinic phase)”.

 Point 2: transition electron microscopy (TEM) …

 Response 2: transmission electron microscope (TEM) is modified into “transmission electron microscopy (TEM)”

 Point 3: Please write uniformly “Zn” or “zinc” throughout the manuscript.

 Response 3: “Zinc” is changed into “Zn” throughout the manuscript.

 Point 4: p. 3, l. 102: “… Table 1 …”

 Response 4: “table 1” was changed into “Table 1”

 Point 5: p. 4, l. 116: “… the decrease of …”

 Response 5: “the decrease the” was changed into “the decrease of”

 Point 6: Fig. 3: The particle size distribution is hardly visible.

 Response 6: The particle size distribution in Fig. 3 is modified to be more distinctly visible.

 Point 7: Fig.s 4 and 11: “Kinetic energy …”

 Response 7: “Knetic” was changed into “Kinetic” in Fig. 4 and Fig. 11.

 Point 8: p. 7, l. 190: “… higher than 0.6, catalytic benzene conversion …”

 Response 8: “the catalytic benzene conversion” was changed into “catalytic activity towards benzene conversion”

 Point 9: p. 9, l. 243: “… Table 2. …”

 Response 9: “table 2” was changed into “Table 2”

 Point 10: Fig. 10: Please check the figure caption. It should be mentioned that ZrO2 is also part of the catalyst system.

 Response 10: “unsupported Ru-Zn (0.60) catalyst before (a, b) and after (c, d) catalytic experiments” was changed into “unsupported Ru-Zn (0.60) catalyst with m-ZrO2 as dispersant before (a, b) and after (c, d) catalytic experiments”.

 Point 11: p. 12, l. 322: “… highest cyclohexene yield of …”

 Response 11: “the highest cyclohexene of” was changed into “the highest cyclohexene yield of”.

 Point 12: Scheme 1: Please explain the step after NaOH addition briefly.

 Response 12: A sentence “Ru(OH)3 could be generated when the NaOH was mixed with the calcined sample in the autoclave during the temperature rising procedure.” was added in chapter 3.2.

 Point 13: What is the role of ZnSO4? Please explain this anywhere in the manuscript.

 Response 13: ZnSO4 is added to react with ZnO to form (Zn(OH)2)3(ZnSO4)(H2O)3, which is of paramount importance for cyclohexene synthesis. And this is mentioned in “Results” Paragraph 1, line 94.

 Point 14: p. 14, l. 383: “… until the filtrate became neutral and …”

 Response 14: “become neutralization” was changed into “became neutral”.
